# Novel Real-Time Compensation Method for Machine Tool's Ball Screw Thermal Error

**Ren Rong, Huicheng Zhou, Yubin Huang \*, Jianzhong Yang and Hua Xiang**

National Numerical Control System Engineering Research Center, Huazhong University of Science and Technology, Luoyu Road 1037, Wuhan 430074, China
\*  Correspondence: 2021512009@hust.edu.cn

**Abstract:** The real-time compensation of thermal error in ball screws is an effective means to improve the accuracy of machining tools. However, the trade-off between robustness and computational efficiency of existing ball screw thermal error models is complicated and not conducive to practical, high-precision, real-time error compensation. Focusing on this problem, we propose an iterative prediction model of screw thermal error based on a finite difference equation. By assuming an approximately linear relationship between heat generation and the ball screw's convection power and feed speed, a simplified and more efficient identification of physical parameters needed for the iterative model is achieved. The proposed method is integrated with a three-axis drilling and tapping machine powered by an HNC–848D controller. A test piece machine using the proposed real-time thermal error compensation method exhibited a maximum machining error of 13 μm, compared to the 71 μm of an uncompensated specimen. The proposed method is demonstrated to improve machining accuracy, especially in the X- and Y- axes, and overcome the limitations of traditional thermal error prediction models.

**Keywords:** machine tool; thermal error; compensation; ball screw feed system

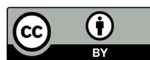

## 1. Introduction

In precision machining, the contribution of thermal error to total machine tool error can reach 40–70% [1,2]. Fortunately, the thermal error of the machine tool can be effectively controlled using materials with low thermal expansion coefficients [3], as well as through structural design optimization [4] and cooling [5]. However, such measures are expensive and, therefore, impractical for most machine tools. For example, the drilling machines commonly used in the 3C (Computers, Communications, and Consumer Electronics) industry utilize a semi-closed control loop, and the thermal error caused by the thermal elongation of the ball screw during processing can significantly reduce positioning accuracy. Therefore, the mechanisms for improving machining accuracy through thermal error compensation technology have been widely researched [6].

For the past few decades, data mining algorithms such as multivariate linear regression [7], neural networks [8], support vector machines [9], and gray systems [10] have been widely used in the modeling of machine tool thermal errors. However, the robustness of the big data model requires suitable heat-sensitive point selection [11]. However, the temperature measurement of moving parts, such as the screw and screw nut, is challenging [12], which has limited the effectiveness of such classic methods.

A potential alternative for thermal error prediction is the solution of the thermal equilibrium equation. However, traditionally, this method exhibits low computational efficiency [13] and challenges in parameter identification [14]. Significant research is being done to overcome these obstacles. Zapłata [15] uses partial differential equations to determine the thermal error model of the ball screw feed system and compensate for it. Liu [16] uses the GA–SRW neural network to construct a thermal error model of the x-axis lead

screw of worm gear machines for error compensation. Meanwhile, Wang [17] has established a quasi-static model and a thermodynamic model for the thermal characteristics of the ball screw and proposes a calculation scheme considering the dynamic changes of boundary conditions. Through comprehensive analysis of the influence of various factors, Ma [18] has determined the mechanism of thermal error in a linear axis fixed on each end (fixed-fixed installation). Chen [19] considers the ball screw feed system as a two-dimensional heat transfer system and has analyzed and predicted its thermal error. Liu [20] proposes a thermal error model of the feed shaft based on the heat transfer mechanism and investigates the reliability of the model based on the deep belief network (DBN) and the Monte Carlo method. Li [14], meanwhile, uses the finite difference method to simulate the thermal elongation of the feed system under various working conditions and optimizes boundary conditions using the response surface method. Mareš [21] directly integrates the thermal error model established using the transfer function into the control system of the machining center to realize real-time compensation of thermal error.

There are few existing studies discussing the trade-off of computing power and the robustness of the thermal error prediction model. In this study, we propose a thermal error compensation method based on a finite difference iterative equation, using the screw of a three-axis drilling machine as the object. The paper is structured as follows. Section 2 presents a broad view of the iterative finite difference equation and modeling method for a single-ended fixed ball screw feed system, while Section 3 presents the detailed derivation of the thermal error equations. Section 4 proposes and evaluates a parameter identification method and introduces the thermal error compensation implementation and procedure. Finally, in Section 5, the compensation process is fully applied, and the effectiveness of the proposed positioning thermal error compensation method is verified through simple measurements, as well as by comparing the machining accuracy of a test workpiece with and without compensation.

## 2. Thermal Equilibrium Analysis

A three-axis drilling machine (powered by an HNC–848 controller board, Huazhong CNC, China) adopting a typical semi-closed control loop is used as the research object in this study. A 3-D model and picture of the machine are shown in Figure 1. The X-, Y-, and Z-axes are driven by a ball screw feed system, fixed on only one end (one-ended installation). Therefore, the position accuracy of the linear axis is highly sensitive to the screw's thermal error.

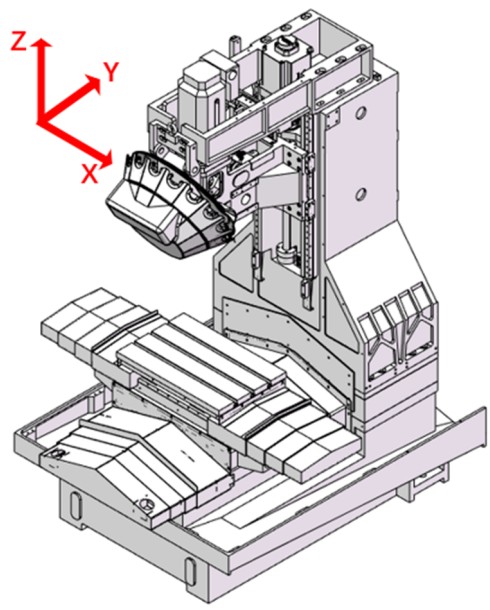
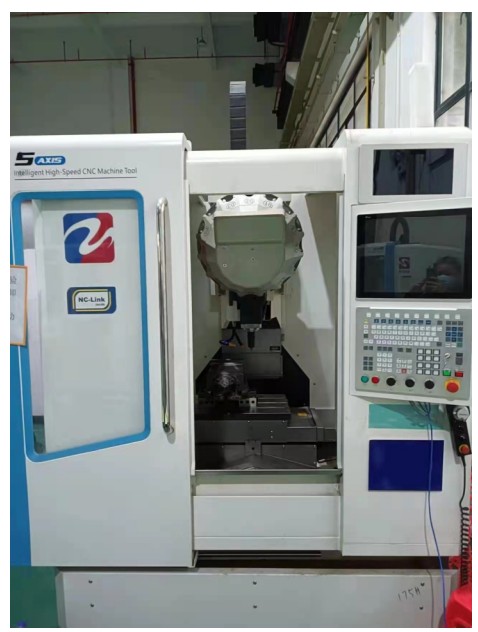

**Figure 1.** Three-axis drilling machine.

## 2.1. Thermal Analysis of Ball Screw Feed System

The structure, heat source, heat conduction, and heat convection of the target ball screw feed system are shown in Figure 2.

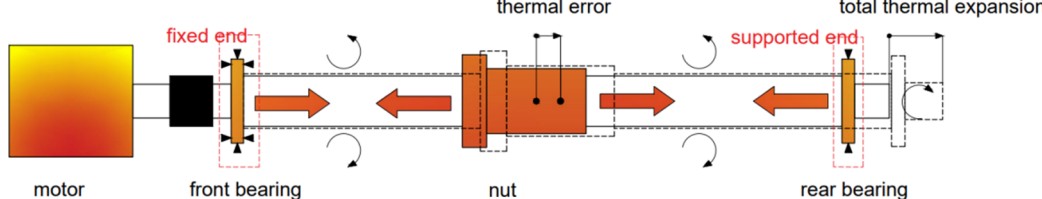

**Figure 2.** Schematic of thermal analysis of ball screw feed system.

The main heat source of the ball screw feed system is the friction in the nut and front bearing. The elastic motor coupling can be regarded as a heat insulator, owing to its large thermal resistance. In addition, the following assumptions simplify the thermal analysis of the ball screw feed system:

1. The thermal boundary at the connection between the fixed end of the screw and the elastic coupling can be regarded as adiabatic. Thus, heat conduction from the motor to the screw can be ignored.
2. Heat generation due to friction in the free-end bearing is negligible and can be ignored.
3. Only the axial heat conduction of the screw is considered.

Then, as shown in Figure 3, the screw can be divided into five sections:

1. Fixed end – This is the section that is in contact with the elastic coupling. The interface between the screw and elastic coupling is considered adiabatic.
2. Fixed bearing – This is the section that is in contact with the fixed-end bearing. Heat generated by the bearing enters the screw here.
3. Nut – The section that is in contact with the ball screw nut. Heat generated by the nut enters the screw here.
4. Free end – The section that is in contact with the free-end bearing.
5. Air convection section – All parts of the screw not identified above. These dissipate the heat of the screw through convection with the air.

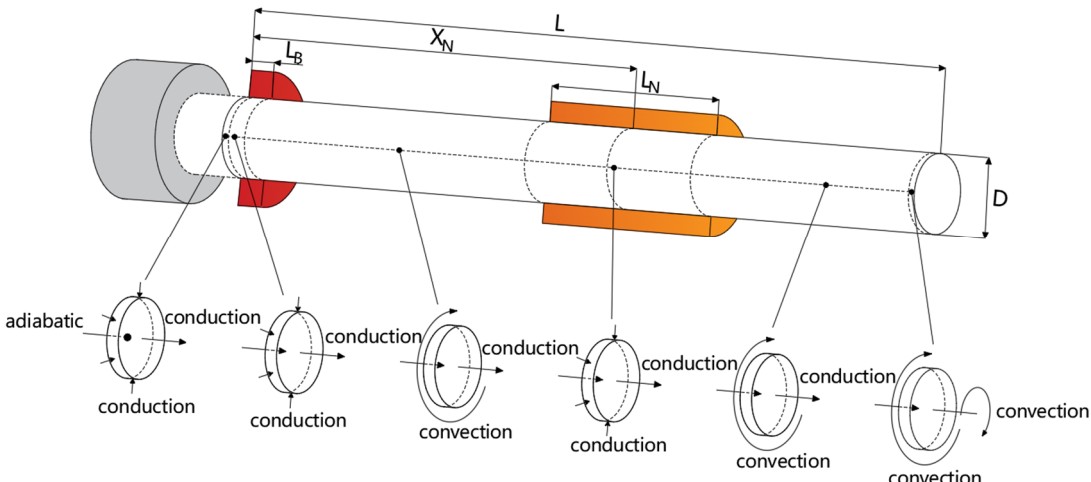

**Figure 3.** Typical sections of the one-end installation screw.

*2.2. Components of Screw Thermal Error*

The thermal error of the ball screw feed system caused by the ambient temperature $TE_a$ can be calculated from a linear equation [22]:

$$TE_a = \alpha \times \Delta T_a \times X_N ,$$
(1)

where $\alpha$ is the expansion coefficient of the screw material, $\Delta T_a$ is the change of ambient temperature, and $X_N$ is the current position of the axis.

The other part of the screw's thermal error is caused by the feed movement of the ball screw feed system, denoted by $TE_m$. The total thermal error of the screw can be expressed as:

$$TE_{total} = TE_a + TE_m$$
(2)

**3. Screw Thermal Error Due to Feed Movement**

The thermal error of the screw due to the feed movement is calculated according to the process illustrated in Figure 4. The algorithm is implemented by matlab in the parameter identification process and implemented in the HNC-848 CNC system by C++ in the error compensation process. An analysis of the thermal boundary of the screw is conducted under the following assumptions:

1. Heat generated by the bearing in the fixed end enters the screw from the face in contact with the bearing.
2. Heat generated by the nut enters the screw from the surface in contact with the nut.
3. Heat convection between the screw and the air occurs on all free surfaces of the screw.

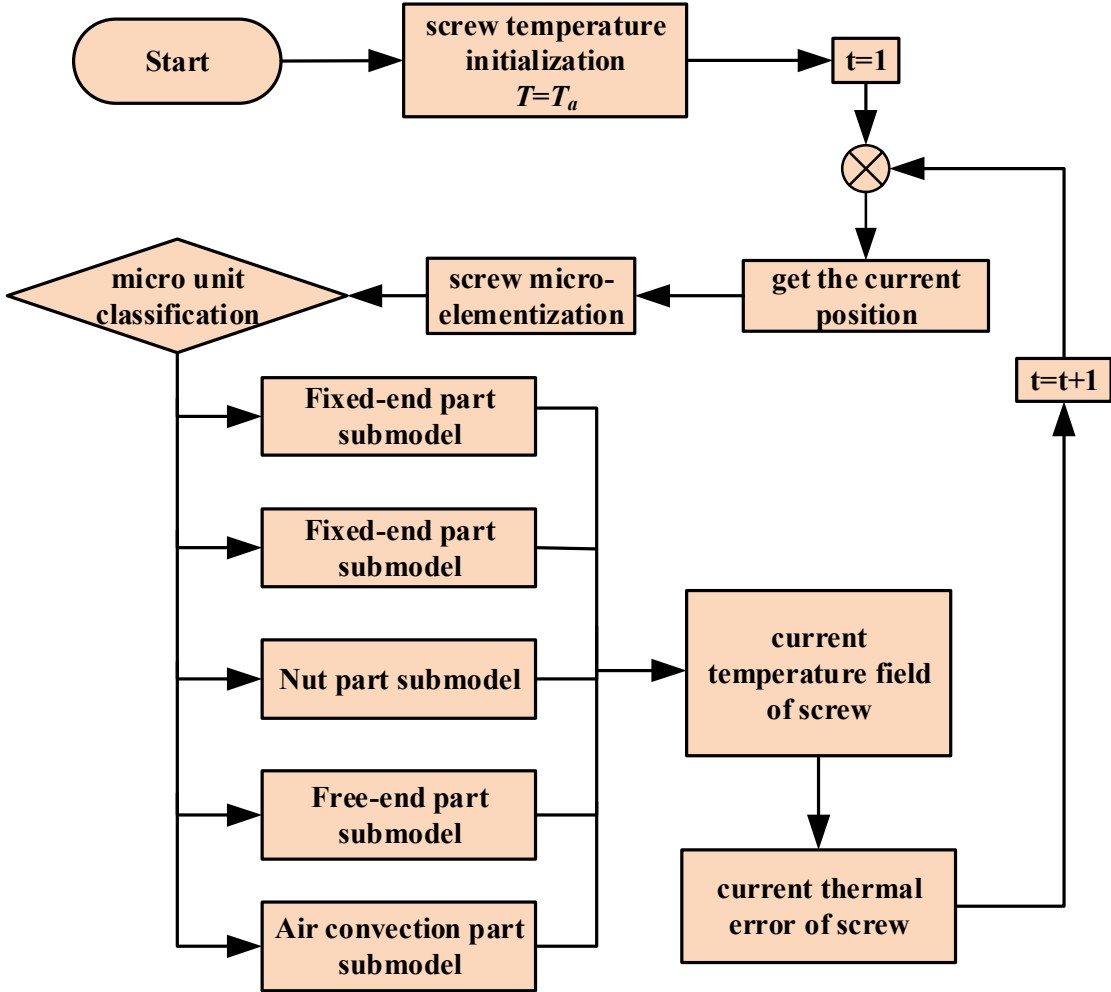

**Figure 4.** Flowchart of screw thermal error prediction.

Under those assumptions, the screw is simplified to a one-dimensional thin rod. According to Fourier's law of thermal conductivity:

$$Q(x,t) = -\lambda \frac{\partial T(x,t)}{\partial x},\tag{3}$$

where $Q(x,t)$ is the heat flow through the cross-section $x$ at time $t$, $T(x,t)$ is the temperature of the cross-section $x$ at time $t$, and $\lambda$ is the thermal conductivity of the screw material.

### 3.1. Derivation of Screw Thermal Distribution

The position of the screw nut varies according to instructions from the controller during operation and thus cannot be solved analytically. Since this affects the temperature field, in this paper, the temperature field is iteratively solved by the finite difference method. First, the screw of total length $L$ is divided into $N$ micro-units of length of $dx$ with uniform internal temperature:

$$N = \frac{L}{dx}.\tag{4}$$

To account for the heat conduction between the left face of the micro-unit and the adjacent micro-unit, the fixed-end micro-unit ($n = 1$) is differentiated from the others. The fixed-end face is regarded as adiabatic; thus, at time $t$, the heat flowing into the $n$-th micro-unit from the left end ($Q^{n,t}_{left}$) can be expressed as:



$$Q_{left}^{n,t} = \begin{cases} 0 \, (n = 1) \\ -\lambda \dfrac{T_n^t - T_{n-1}^t}{dx} A_s \, (\text{else}) \end{cases} \tag{5}$$

where $A_s = \frac{\pi D^2}{4}$ is the approximate cross-sectional area of the screw.

Similarly, for the micro-unit at the free-end face, the heat flow $Q^{n,t}_{right}$ out of the $n$-th micro-unit from the right end at time $t$ can be expressed as:

$$Q_{right}^{n,t} = \begin{cases} h(T_n^t - T_a)A_s \, (n = N) \\ -\lambda \dfrac{T_{n+1}^t - T_n^t}{dx} A_s \, (\text{else}) \end{cases} \tag{6}$$

where $h$ is the heat convection coefficient between the screw and the air, and $T_a$ is the ambient temperature.

Heat transfer on the cylindrical surface of the micro-unit occurs in three sections: air convection, nut, and fixed-end bearing. The heat flowing into the $n$-th micro-unit from the circular face at time $t$ ($Q^{n,t}_{circle}$) can be expressed as:

$$Q_{circle}^{n,t} = \begin{cases} -h(T_n^t - T_a)A_C \, (\text{Air convection part}) \\ \dfrac{Q_N^t}{L_N} g dx \, (\text{Nut part}) \\ \dfrac{Q_B^t}{L_B} g dx \, (\text{Fixed end bearing part}) \end{cases} \tag{7}$$

where $Q_N^t$ is the heat flow into the screw from the screw nut at time $t$, $L_N$ is the length of the screw nut, $Q_B^t$ is the heat flow into the screw from the fixed-end bearing, and $L_B$ is the length of the fixed-end bearing.

The heat stored in the $n$-th micro-unit at time $t$ can be expressed as:

$$Q_{in}^{n,t} = \rho c A_s dx \frac{T_n^{t+dt} - T_n^t}{dt} \tag{8}$$

where $\rho$ is the density of the screw material, $c$ is the specific heat capacity of the screw material, and $dt$ is the iterative calculation cycle time.

According to the principle of heat conservation:

$$Q_{left}^{n,t} + Q_{circle}^{n,t} = Q_{right}^{n,t} + Q_{in}^{n,t}. \tag{9}$$

Combining the above formulas, the iterative calculation formula for predicting the temperature field of the screw at time $t+dt$, based on the temperature at time $t$, can be expressed as follows:

$$T_n^{t+dt} = \begin{cases} T_n^t + \dfrac{\lambda dt}{\rho c(dx)^2}(T_{n+1}^t - T_n^t) + \dfrac{4dt}{\pi D^2 \rho c}(\dfrac{Q_B^t}{L_B})(n=1) \\[4mm] T_n^t + \dfrac{\lambda dt}{\rho c(dx)^2}(T_{n+1}^t + T_{n-1}^t - 2T_n^t) + \dfrac{4dt}{\pi D^2 \rho c}(\dfrac{Q_B^t}{L_B})(1 < n \le \dfrac{L_B}{dx}) \\[4mm] T_n^t + \dfrac{\lambda dt}{\rho c(dx)^2}(T_{n+1}^t + T_{n-1}^t - 2T_n^t) + \dfrac{4dt}{\pi D^2 \rho c}(\dfrac{Q_N^t}{L_N})(\dfrac{X_N^t - \dfrac{L_N}{2}}{dx} < n \le \dfrac{X_N^t + \dfrac{L_N}{2}}{dx}) \\[4mm] T_n^t + \dfrac{\lambda dt}{\rho c(dx)^2}(T_{n-1}^t - T_n^t) - (\dfrac{4hdt}{\rho cD}(1 + \dfrac{D}{4dx}))(T_n^t - T_a)(n=N) \\[4mm] T_n^t + \dfrac{\lambda dt}{\rho c(dx)^2}(T_{n+1}^t + T_{n-1}^t - 2T_n^t) - (\dfrac{4hdt}{\rho cD})(T_n^t - T_a)(\text{else}) \end{cases} \tag{10}$$

Assuming that the temperature of the screw is equal to the ambient temperature when the machine tool starts:

$$T_n^0 = T_a.$$

Thus, the temperature distribution of the screw at any time can be calculated iteratively through the above formula.

### 3.2. Prediction of Screw Thermal Error

The thermal expansion of the $n$-th micro-unit at time $t$ can be expressed as:

$$E_n^t = \alpha(T_n^t - T_a) g dx, \tag{11}$$

where $\alpha$ is the coefficient of thermal expansion.

We define $TE_m(x,t)$ as the thermal error of the screw at position $x$ at time $t$. Then, for the fixed end of the screw:

$$TE_m(0,t) = 0. \tag{12}$$

Thus, the thermal error of the $n$-th micro-unit is:

$$TE_m(n g dx, t) = \sum_{i=1}^{n} E_i^t (1 \le n \le N). \tag{13}$$

The thermal error of the screw at position $x$ and time $t$ can then be solved according to the following equation:

$$TE_m(x,t) = TE_m(m g dx, t) + \frac{x - m g dx}{dx} E_{m+1}^t (m g dx < x < (m+1) g dx) \tag{14}$$

where $P_f$ denotes the coordinates of the analyzed axis when the screw nut is at the fixed end of the screw. Thus, $TE(P_x,t)$ is the thermal error when the control coordinate is $P_x$ at time $t$. It follows that:

For $P_f > 0$, the thermal error of $P_x$ is:

$$TE(P_x,t) = -TE_m(P_f - P_x, t) - TE_a \tag{15}$$

For $P_f < 0$, the thermal error of $P_x$ is:

$$TE(P_x,t) = TE_m(P_x - P_f, t) + TE_a \tag{16}$$

### 4. Parameter Identification of Simplified Model and Application of Compensation

*4.1. Simplification of Parameter Expressions*

Most parameters (e.g., material properties) required for the screw's thermal error prediction model could be determined from the screw and machine manuals (Table 1), but some required calculation. The convection heat transfer coefficient *h*, the heat flow from the fixed-end bearing $Q_B$, and the heat flow from the ball screw nut $Q_N$ are difficult to determine through theoretical calculation. However, since these three parameters are positively correlated with the feed speed [15], they could be simplified into the following expressions:

$$h = k_1 v + k_2 \tag{17}$$

$$Q_B = k_3 v \tag{18}$$

$$Q_N = k_4 v \tag{19}$$

Here, $k_1$, $k_2$, $k_3$, and $k_4$ are parameters to be identified, and *v* is the feed speed.

**Table 1.** Parameters determined from the screw and machine manuals.

| C (J/kg·°C) | ρ (kg/m³) | L (m) | D (m) | $L_N$ (m) | $L_B$ (m) | α (μm/m·°C) | dx (m) | dt (s) | N |
|---|---|---|---|---|---|---|---|---|---|
| 448 | 7800 | 0.8 | 0.034 | 0.212 | 0.03 | 11.7 | 0.01 | 0.1 | 80 |

*4.2. Parameter Identification Method*

Identification of the above proportional parameters can be considered an optimization problem, as follows:

$$\min : Y(k_1, k_2, k_3, k_4, \lambda, P_f) = \sum (TE_{predict} - TE_{real})^2$$
$$s.t. \begin{cases} k_1 > 0 \\ k_2 > 0 \\ k_3 > 0 \\ k_4 > 0 \\ \lambda > 0 \end{cases}, \tag{20}$$

where $TE_{predict}$ and $TE_{real}$ are the predicted and measured positioning thermal errors of the screw feed system, respectively.

After obtaining measured positioning thermal error $TE_{real}$ (as described in the next section), the above parameters are identified by sequential quadratic programming (SQP) [23]. As a mature algorithm, the principle is not discussed in this paper. The solution of the optimization equation in the process of parameter identification is completed by matlab.

*4.3. Application of Screw Thermal Error Compensation*

Based on the above model, a function for the compensation of the screw thermal error was integrated into the HNC-848 CNC system by C++ programming, as illustrated in Figure 5. The iterative calculation of the thermal error of the screw was carried out in a weak real-time environment. After the full-length thermal error array of the screw is obtained, the thermal error compensation command is calculated in real time according to the interpolation of the current position in the real-time kernel. After the thermal error

compensation function is integrated, the thermal error compensation process is shown in Figure 6.

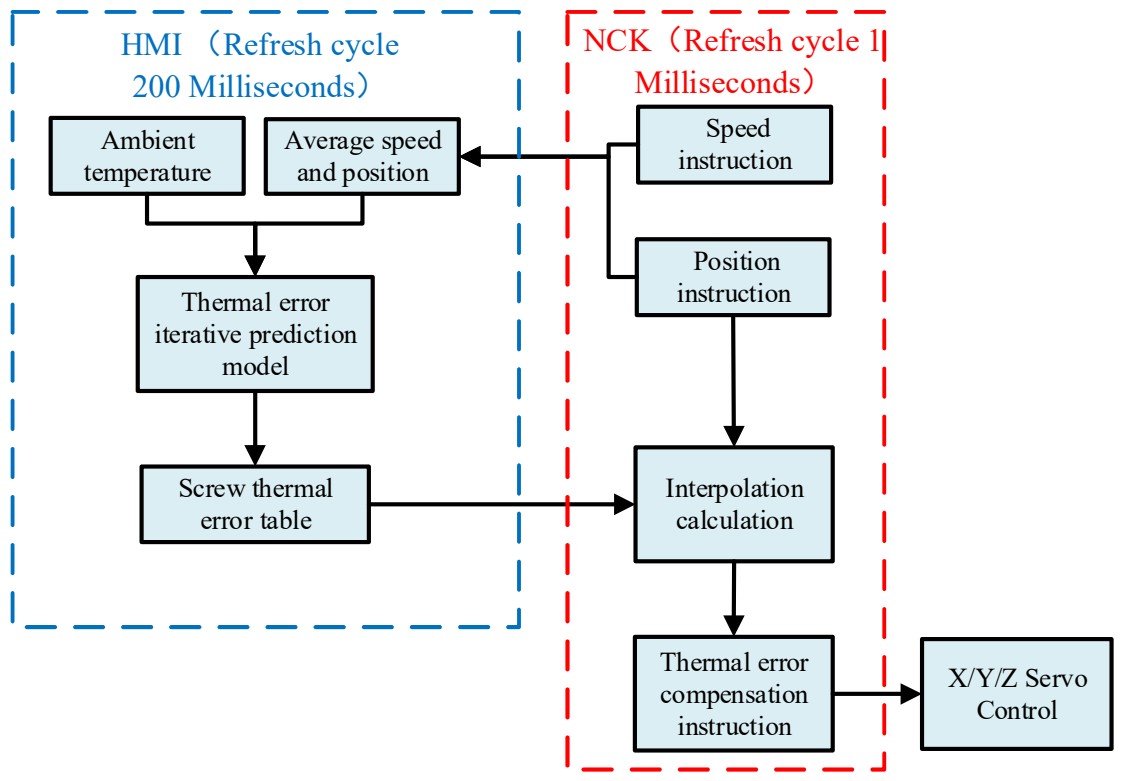

**Figure 5.** Implementation of the thermal error compensation module in the CNC system.

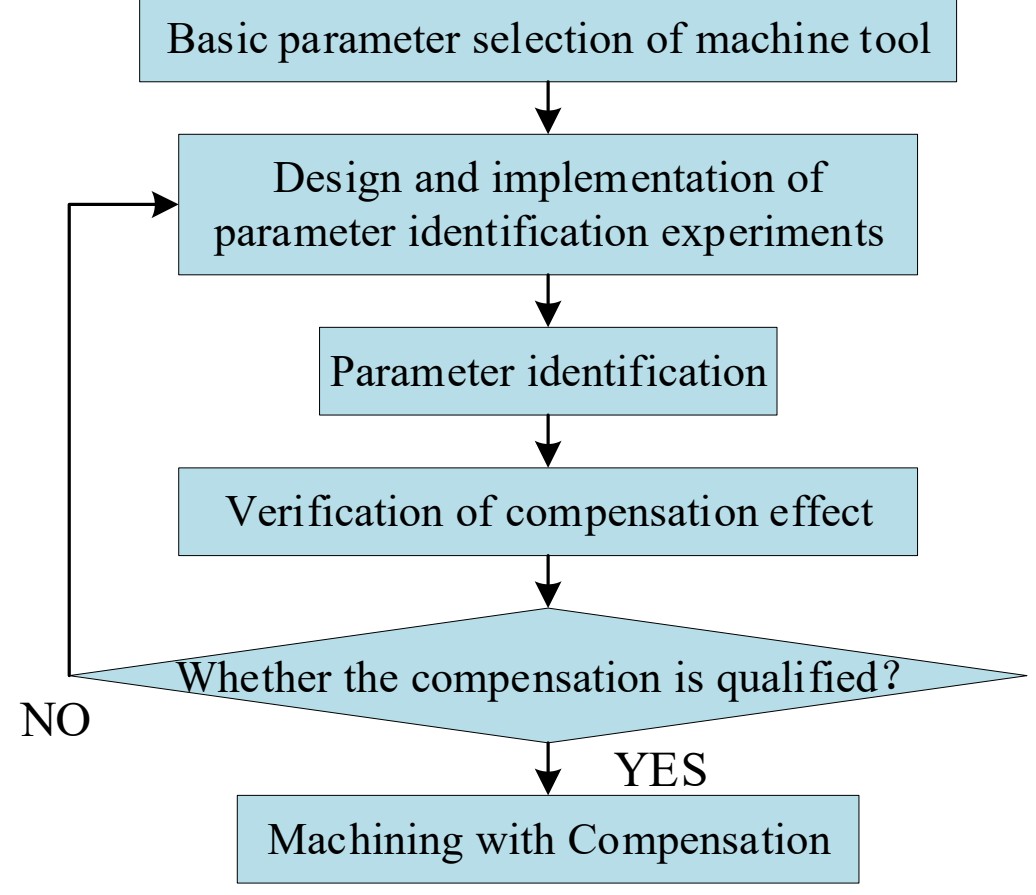

**Figure 6.** Thermal error compensation process.

## 5. Verification Experiment

### 5.1. Parameter Identification Experiment

The actual screw thermal error required for parameter identification was measured by a multi-degree laser interferometer (XM-60, Renishaw). The measurement setup is shown in Figure 7. The XM-60 measurement process is shown in Figure 8. To simulate the processing-shutdown-processing procedure typical of a practical industrial production scenario and, thereby, improve the robustness of the model, multiple sets of loading experiments with different simulation loads were conducted. These are summarized in Table 2.

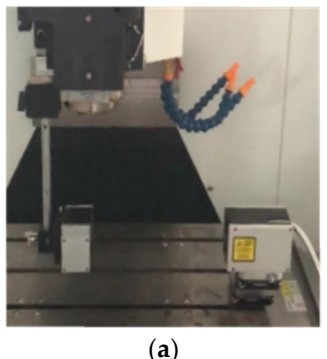 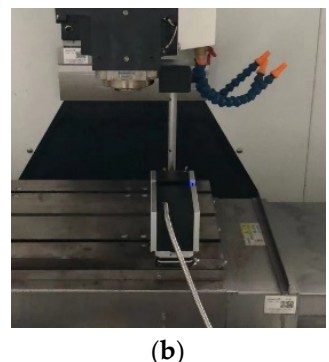 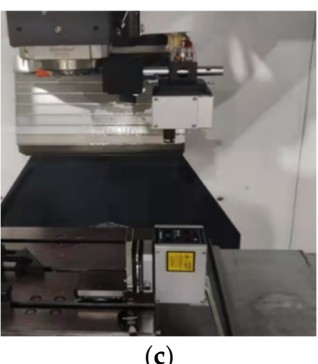

(**a**) (**b**) (**c**)

**Figure 7.** Positioning thermal error measurement. (**a**) X-axis. (**b**) Y-axis. (**c**) Z-axis.

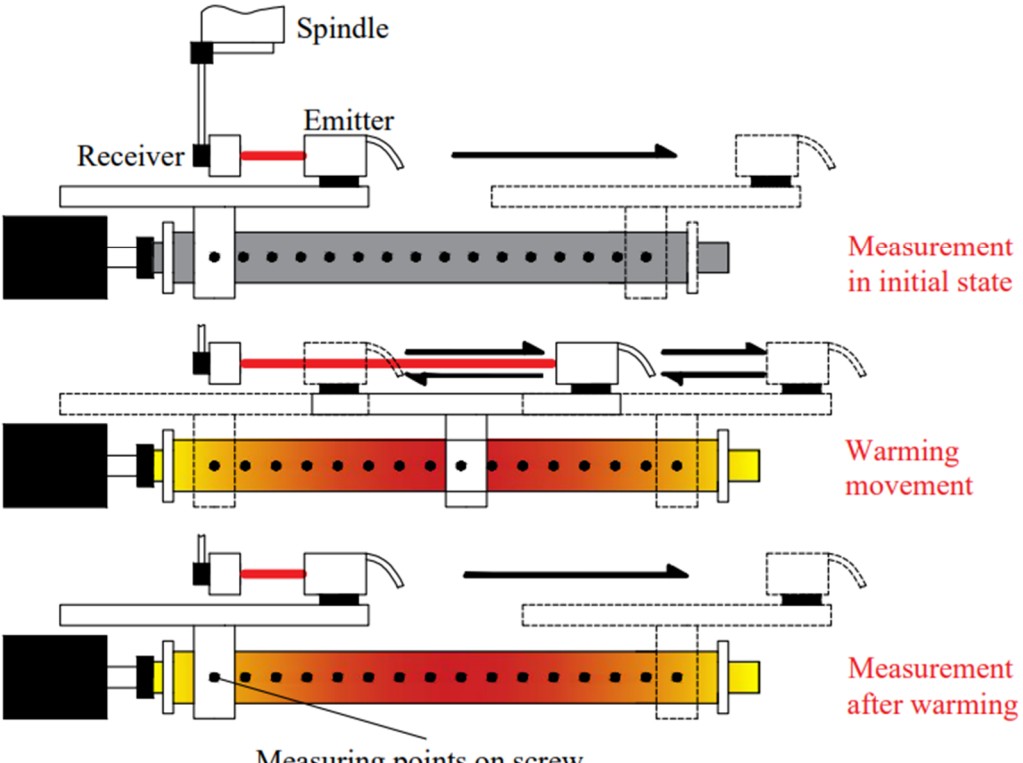

**Figure 8.** XM–60 measurement process (x-axis as an example).

**Table 2.** Simulation loading method of parameter identification.

| Condition: | State 1 | State 2 | State 3 | Warming Feed Speed (mm/min) |
|---|---|---|---|---|
| 1 | Warming 84 min | Cooling: 42 min | Warming 42 min | 3000 |
| 2 | | | | 6000 |
| 3 | | | | 9000 |

The parameter values of each axis, resulting from the optimization exercise, are shown in Table 3. The measured thermal error and residual error of the thermal error prediction model are shown in Figures 9–11 for each respective axis.

**Table 3.** Identified parameters of each axis.

| Axis | $k_1$ | $k_2$ | $k_3$ | $k_4$ | $\lambda$ | $P_f$ |
|---|---|---|---|---|---|---|
| X | $0.911 \times 10^{-3}$ | 26.034 | $0.904 \times 10^{-3}$ | $0.911 \times 10^{-3}$ | 566.3 | −0.141 |
| Y | $0.703 \times 10^{-3}$ | 16.746 | $0.997 \times 10^{-3}$ | $1.76 \times 10^{-3}$ | 278.1 | −0.655 |
| Z | $2.114 \times 10^{-3}$ | 24.736 | $0.041 \times 10^{-3}$ | $4.103 \times 10^{-3}$ | 308.8 | 0.151 |

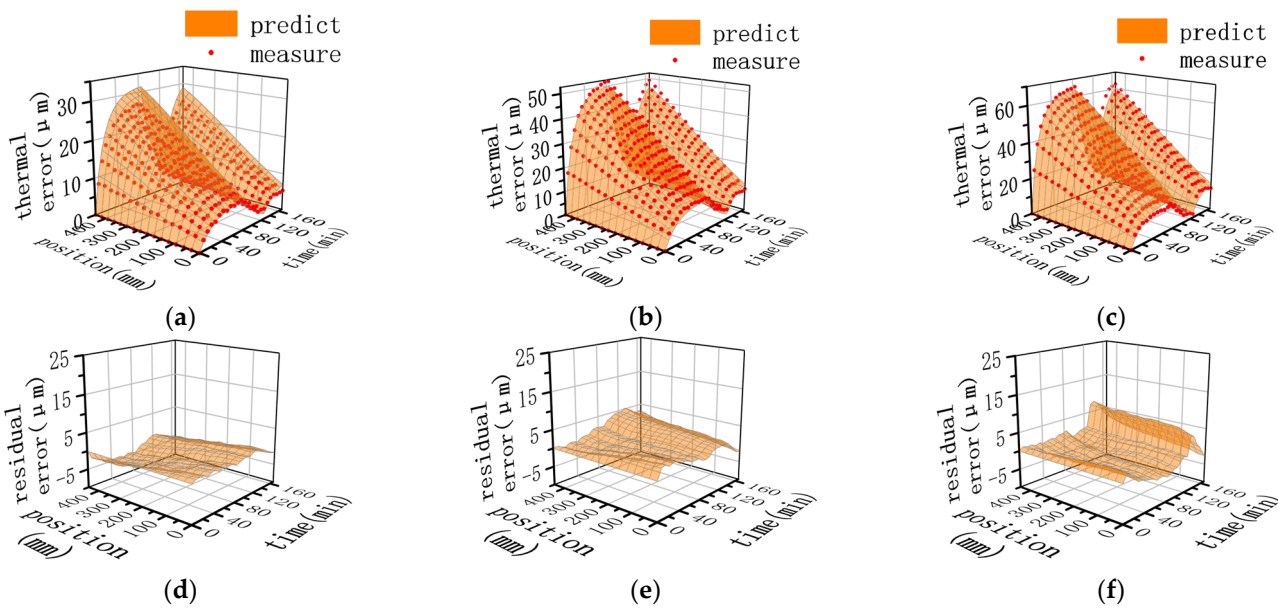

**Figure 9.** X-axis error results. (**a**) Measured thermal error in condition 1. (**b**) Measured thermal error in condition 2. (**c**) Measured thermal error in condition 3. (**d**) Residual error in condition 1. (**e**) Residual error in condition 2. (**f**) Residual error in condition 3.

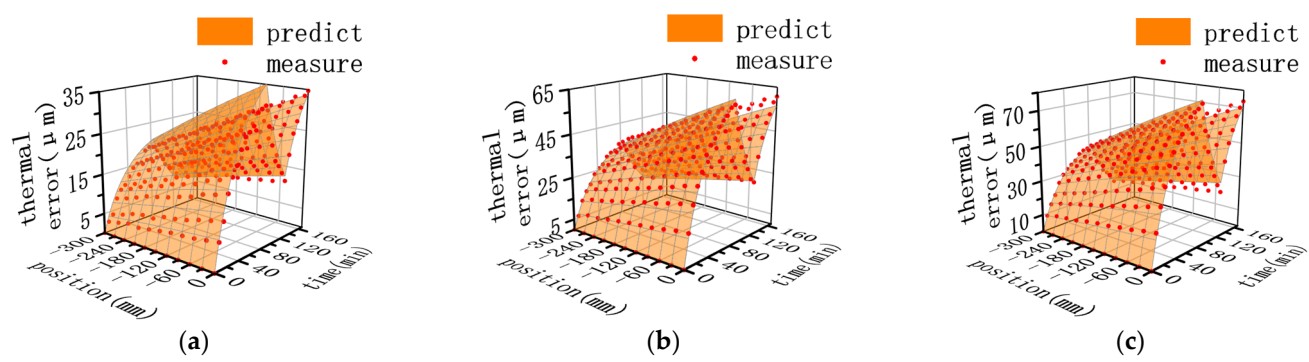

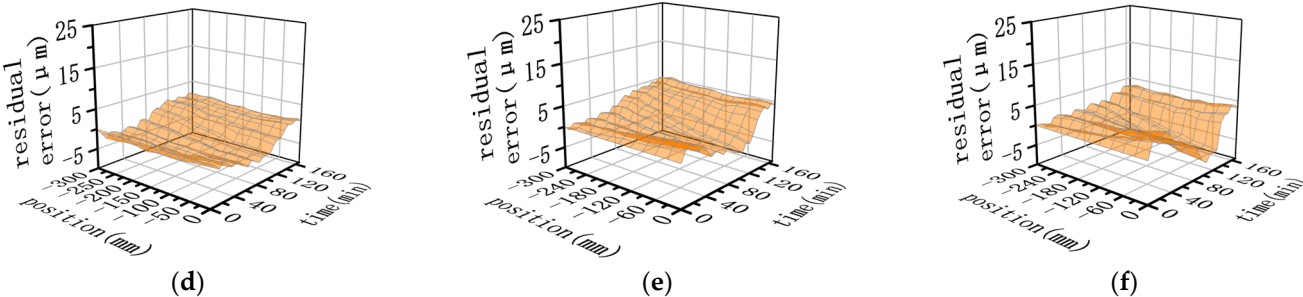

**Figure 10.** Y-axis error results. (**a**) Measured thermal error in Condition 1. (**b**) Measured thermal error in Condition 2. (**c**) Measured thermal error in Condition 3. (**d**) Residual error in Condition 1. (**e**) Residual error in Condition 2. (**f**) Residual error in Condition 3.

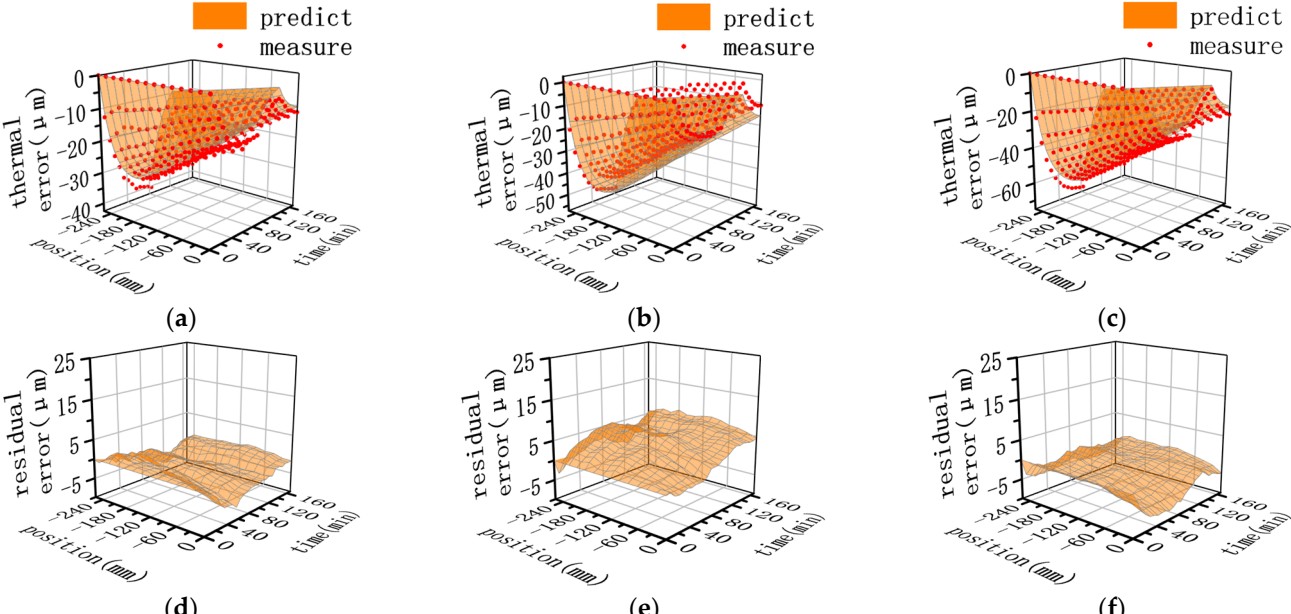

**Figure 11.** Z-axis error results. (**a**) Measured thermal error in Condition 1. (**b**) Measured thermal error in Condition 2. (**c**) Measured thermal error in Condition 3. (**d**) Residual error in Condition 1. (**e**) Residual error in Condition 2. (**f**) Residual error in Condition 3.

The residual error of the thermal error prediction model is within 15% of the measured thermal error P–V value. Thus, the parameter identification algorithm has sufficient fitting accuracy for the purposes of this experiment, and the parameter simplification method proposed in this paper is validated.

### 5.2. Verification of Compensation Method

An experiment was conducted to verify the effectiveness of the proposed thermal error compensation method, following a similar process to that in the parameter identification experiment. The simulation loading conditions used for the verification experiment are shown in Figure 12. In order to verify the repeatability of the compensation effect, three sets of independent thermal error compensation effect verification experiments were carried. The final experimental results are shown in Figure 13.

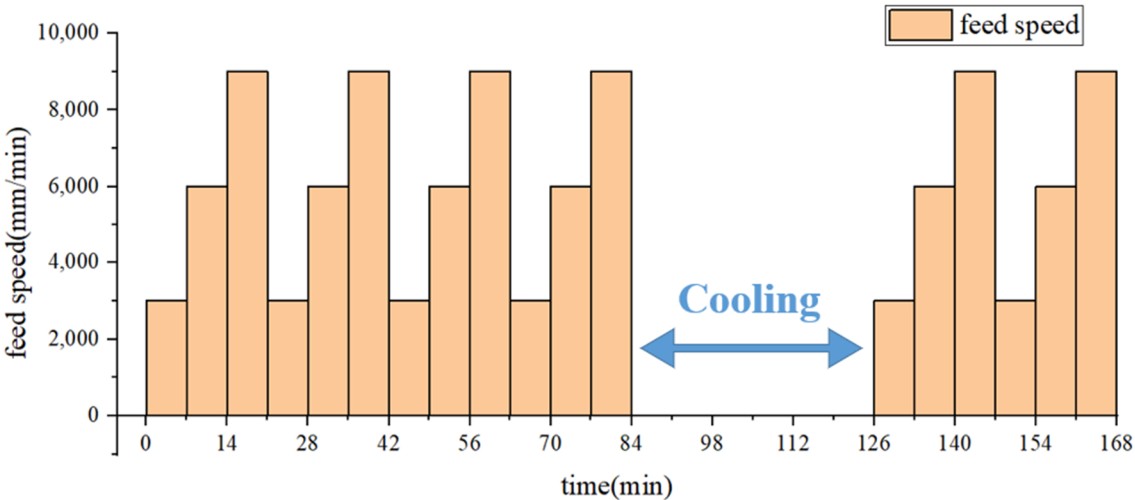

**Figure 12.** Simulated loading for verification experiment.

Before thermal error compensation, the maximum thermal positioning error of the X-, Y-, and Z- axes was 57.2 μm, 57.3 μm, and 56.1 μm, respectively. With thermal error compensation active, the maximum positioning thermal error of three sets of in verification experiments of each axis was within 10 μm. This translates into a position error reduction of approximately 80%, demonstrating the effectiveness of the proposed thermal positioning error compensation method.

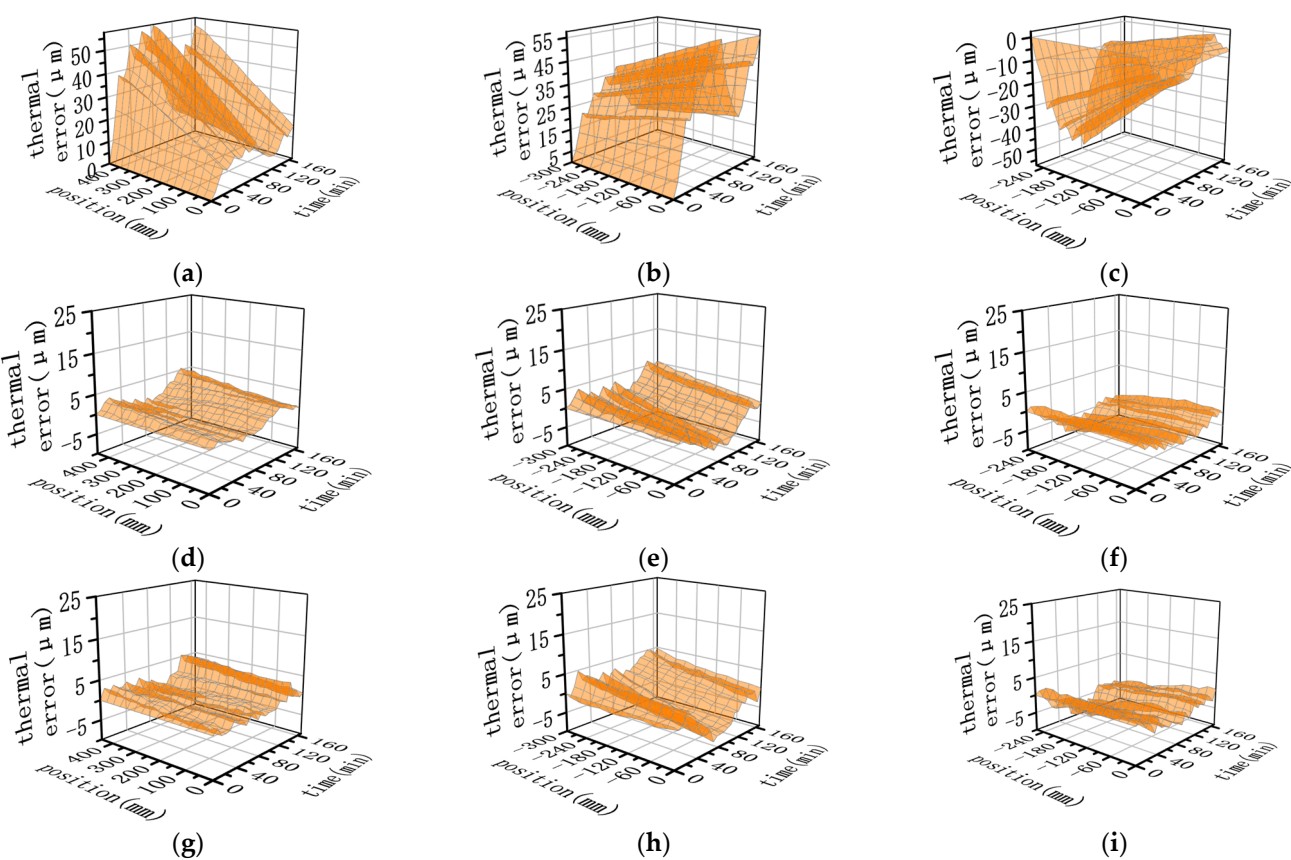

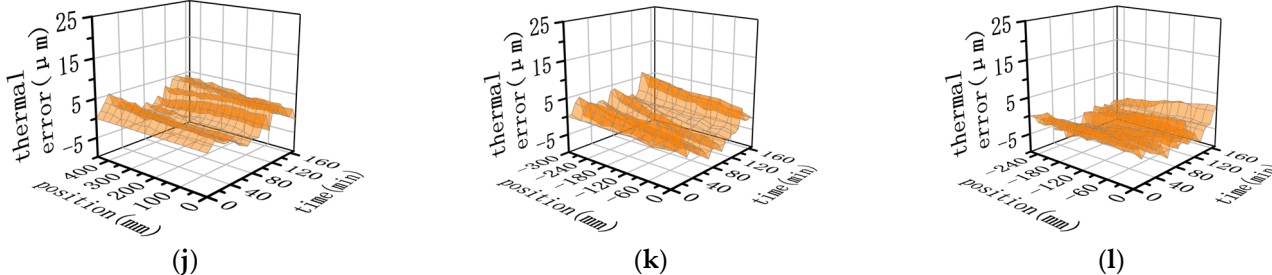

**Figure 13.** Measured positioning thermal error. (**a**) X-axis without compensation. (**b**) Y-axis without compensation. (**c**) Z-axis without compensation. (**d**) X-axis with compensation (1st). (**e**) Y-axis with compensation (1st). (**f**) Z-axis with compensation (1st). (**g**) X-axis with compensation (2nd). (**h**) Y-axis with compensation (2nd). (**i**) Z-axis with compensation (2nd). (**j**) X-axis with compensation (3rd). (**k**) Y-axis with compensation (3rd). (**l**) Z-axis with compensation (3rd).

*5.3. Thermal Error Test Piece Machining*

A novel test piece was designed to evaluate the effect of the proposed machine tool positioning the thermal error compensation method. As shown in Figure 14, the proposed test piece was designed to avoid the influence of tracking errors on its machining accuracy. The machining error is determined by the positioning error of the X-/Y-/Z-axis feed system and the geometric error of the machine tool. The geometric error of the machine tool is regarded as a constant, and the spindle of the machine tool was warmed up fully before machining to avoid the influence of the spindle's thermal error. Thus, any variation in the test piece's machining error with varying feed load is completely determined by the positioning thermal error of the feed system.

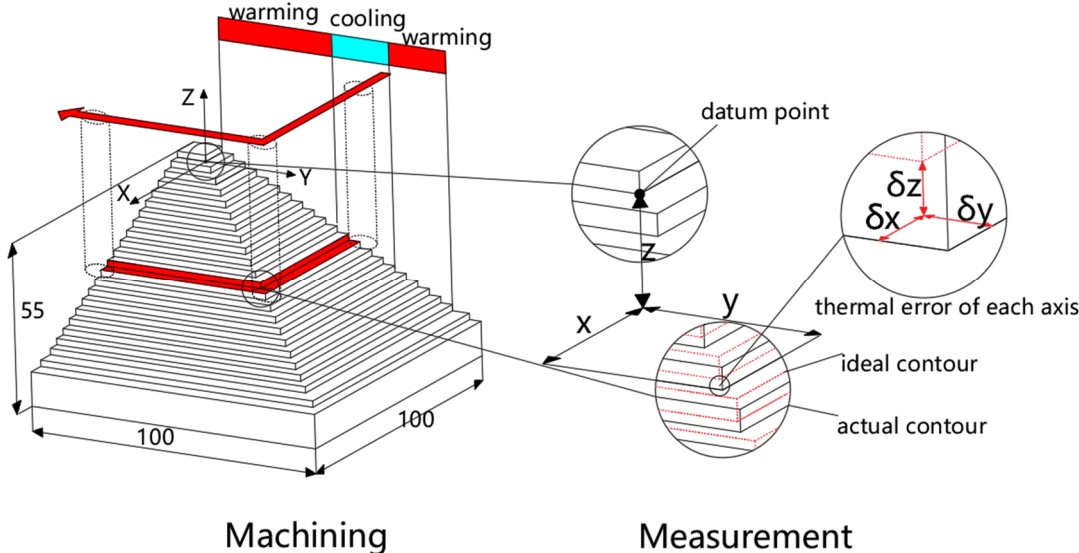

**Figure 14.** Custom test piece for evaluation of positioning thermal error.

The process of machining and measuring the test piece is shown in Figure 15. The machining accuracy of the test piece was measured using a high-precision three-coordinate machine (Renishaw & WENZEL), and the measurement results are shown in Figure 16.

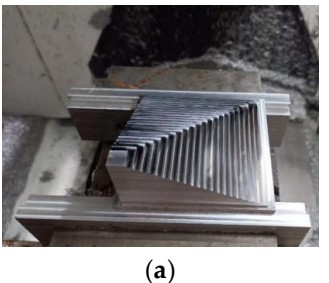

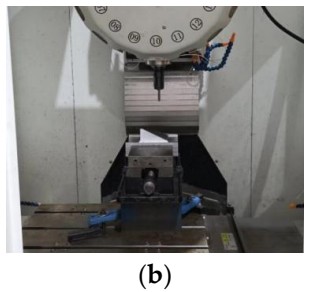

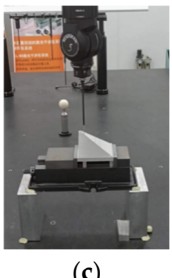

(**a**)    (**b**)    (**c**)

**Figure 15.** Machining and measurement of the test piece. (**a**) Clamping. (**b**) Machining. (**c**) Measurement.

After compensation, the maximum thermal error was reduced by 88.7%, 87.1%, and 56.7% in the X-, Y-, and Z-directions, respectively. Compensation was least effective in the Z-direction, mainly due to the Z-direction machining accuracy being greatly affected by the thermal error of the spindle and the thermal deformation of the column structure. This is also one of the problems to be studied and solved in the future.

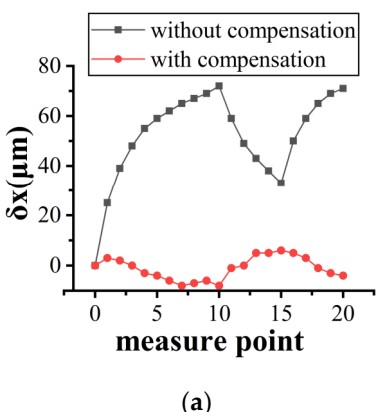

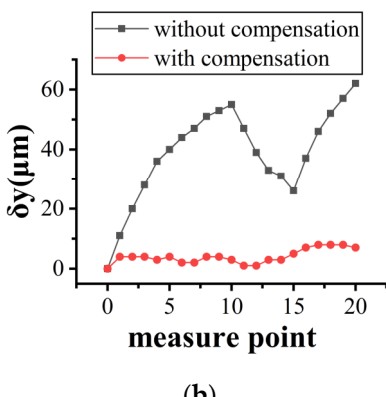

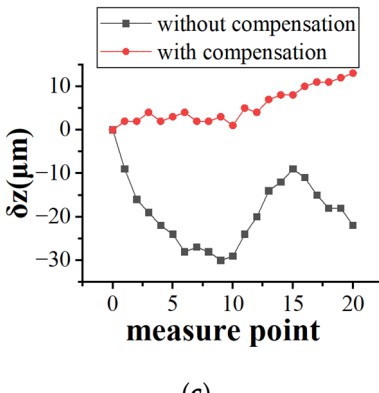

(**a**)    (**b**)    (**c**)

**Figure 16.** Test piece measurement results. (**a**) $\delta x$. (**b**) $\delta y$. (**c**) $\delta z$.

## 6. Conclusions

In this paper, a real-time thermal error compensation method of a ball screw feed system is proposed. A new iterative prediction method of the machine tool's screw thermal error based on a finite difference equation is developed. Compared with the existing method of screw thermal error prediction, the proposed method in this paper has a better calculation efficiency while ensuring accuracy. The real-time compensation function of the screw thermal error based on the above method is integrated into the CNC system.

A simple measurement verification experiment shows that the compensation function can reduce the screw thermal error of the feed axis to within 10 µm. A comparative analysis of a test piece, machined with and without compensation, resulted in a reduction of the maximum machining error from 71 µm to 13 µm after compensation, which verifies the efficacy of the proposed screw thermal error prediction method and compensation function. The proposed model overcomes some of the challenges of traditional error compensation models, including high computational expense and difficult parameter identification.

Currently, our work is based on the no-load or light load of the machine tool. The future research will focus on the influence of complex load on the thermal error of the lead screw and consider the research through the electronic control data in the processing process.

**Author Contributions:** Conceptualization, H.Z. and Y.H.; methodology, H.Z. and R.R; software, R.R.; validation, R.R. and Y.H.; formal analysis, J.Y.; investigation, Y.H. and R.R.; resources, H.X. and J.Y; data curation, H.Z.; writing—original draft preparation, R.R.; writing—review and editing, Y.H.; visualization, R.R.; supervision, H.Z.; project administration, J.Y.; funding acquisition, H.X. All authors have read and agreed to the published version of the manuscript.

**Funding:** This research was funded by Major science and technology project of Hubei Province (2021AAB001) and High Grade CNC System and Servo Motor Project (TC210H03A-04).

**Institutional Review Board Statement:** not applicable

**Informed Consent Statement:** not applicable

**Data Availability Statement:** The data that support the findings of this study are available on request from the corresponding author, [Y. H.] upon reasonable request.

**Conflicts of Interest:** The authors declare no conflict of interest.

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
