# Peer review of "Novel Real-Time Compensation Method for Machine Tool’s Ball Screw Thermal Error"

_applsci, doi:10.3390/app13052833_

Round 1

Reviewer 1 Report

In this paper, the authors proposed an iterative prediction model of screw thermal error to improve the accuracy of machining tools. The proposed model is integrated with a three-axis drilling and tapping machine.

From my point of view there are some aspects to improve:

1.      The structure of the paper is unclear for a good understanding of the reader. Thus the paper structure should be revised in order to clearly present the Material and Methods, and Results in separated sections. The Material and Method section should describe clearly the materials used and the proposed methods. The section Results should shown all the results obtained by applying the proposed method. Thus, Figures 5, 6, 7, 10, 11 and 14 present results and they should be moved to the Results section. 

2. Please explain in the paper, the “Novel real-time compensation method for machine tool’s ball”, specifying what is new.

3.  A statistical analysis of the data should be done. How was verified the repeatability of the data?

4.   What software was used to generate Figure 6-8?

5. What software was used for the implementation of the algorithm presented in the Figure 4 “Flowchart of screw thermal error prediction”?

6.  All the conclusions should be done based o the results. Also, future research directions may also be highlighted.

7.  Are the limitations of this study noted? The limitations of this study should be discussed.

Author Response

Response to Reviewer 1 Comments

Point 1: The structure of the paper is unclear for a good understanding of the reader. Thus the paper structure should be revised in order to clearly present the Material and Methods, and Results in separated sections. The Material and Method section should describe clearly the materials used and the proposed methods. The section Results should shown all the results obtained by applying the proposed method. Thus, Figures 5, 6, 7, 10, 11 and 14 present results and they should be moved to the Results section.

Response 1: Thank you for your suggestion, the structure of the paper has been modified in revised manuscript. In revised manuscript, the paper is structured as follows. Section 2 presents a broad view of the iterative finite difference equation and modeling method for a single-ended fixed ball screw feed system, while section 3 presents the detailed derivation of the thermal error equations. Section 4 proposes and evaluates a parameter identification method, and introduces the thermal error compensation implementation and procedure. Finally, in Section 5, the compensation process is fully applied, and the effectiveness of the proposed positioning thermal error compensation method is verified through simple measurements, as well as by comparing the machining accuracy of a test workpiece with and without compensation.

Point 2: Please explain in the paper, the “Novel real-time compensation method for machine tool’s ball”, specifying what is new.

Response 2: Thank you for your suggestion, it has been supplemented in the conclusion of the revised manuscript as: ” In this paper, a real-time thermal error compensation method of a ball screw feed system is proposed. A new iterative prediction method of machine tool’s screw thermal error based on a finite difference equation is developed. Compared with the existing method of screw thermal error prediction, the proposed method in this paper has better calculation efficiency while ensuring accuracy. The real-time compensation function of screw thermal error based on the above method is integrated into the CNC system. ”.

Point 3: A statistical analysis of the data should be done. How was verified the repeatability of the data?

Response 3: Thanks for your questiuon, it has been modified in the revised manuscript. The reliability of the measurement system can be guaranteed by the Renishaw XM60 laser interferometer. For the data before compensation, due to the difference in the ambient temperature of the experiment itself, the measurement results are not repeatable. In the revised article, for each axis The results of the repeatability experiment after compensation are additionally supplemented, and the experiment proves that the compensation function is repeatable.

The related description has been added into the revised manuscript as: “ In order to verify the repeatability of the compensation effect, three sets of independent thermal error compensation effect verification experiments were carried. ” and the discussion of the compensation effect has been modified as: “With thermal error compensation active, the maximum positioning thermal error of three sets of in verification experiments of each axis was within 10 µm. ”

Point 4: What software was used to generate Figure 6-8?

Response 4: Figs. 6-8 were generated by Origin 2018.

Point 5: What software was used for the implementation of the algorithm presented in the Figure 4 “Flowchart of screw thermal error prediction”?

Response 5: In the stage of parameter identification, it is implemented by matlab programming, and the parameters to be identified are calculated by solving the optimization equation in matlab. In the stage of thermal error compensation, based on the secondary development platform of Huazhong 8 CNC system, it is implemented by C++ programming, and the thermal error compensation value is calculated in real time. In the future, the parameter identification is ready to be applied in the Huazhong 9 CNC system and realized as a product.

Point 6: All the conclusions should be done based o the results. Also, future research directions may also be highlighted.

Response 6: Thank you for your suggestion, the relevant content has been supplemented in the conclusion of the revised manuscript as: ” The future research will focus on the influence of complex load on the thermal error of the lead screw, and consider the research through the electronic control data in the processing process.”.

Point 7: Are the limitations of this study noted? The limitations of this study should be discussed.

Response 7: Thank you for your suggestion, the relevant content has been supplemented in the conclusion of the revised manuscript as: ” Currently,our work is based on the no-load or light load of the machine tool.”.

Reviewer 2 Report

This paper presents a thermal error compensation methodology applied to tracking error reduction in a three-axis drilling CNC machine. In general, the manuscript is well structured, experimentation is adequate to validate the proposal. However, I consider that there are some aspects to consider before a possible publication:

1.       Define T(x,y) in equation 3.

2.       The labels of the axes in figures 6, 7, 8 and 11 are not clear, please improve them.

3.       The lambda constant represents the coefficient of thermal conductivity. Why does this value have to be identified as a variable? What material is the ball screw made of, both in simulation and experimentation? Table 2 presents different lambda values for each axis, if the screw material is the same for each axis, why is there a big difference in values?

4.       Define the main variables involved in experimentation and simulation. Screw material and its coefficient of thermal conductivity, total length L of the screw, number N of micro-units, dx value, ambient temperature, etc.

5.       Is the CNC system used in the experiment open source or reconfigurable? How is the estimated thermal error fed back into the control loop?

Author Response

Response to Reviewer 2 Comments

Point 1: Define T(x,y) in equation 3.

Response 1: Thank you for your suggestion, it has been modified.

Point 2: The labels of the axes in figures 6, 7, 8 and 11 are not clear, please improve them.

Response 2: Thank you for your suggestion, it has been modified.

Point 3: The lambda constant represents the coefficient of thermal conductivity. Why does this value have to be identified as a variable? What material is the ball screw made of, both in simulation and experimentation? Table 2 presents different lambda values for each axis, if the screw material is the same for each axis, why is there a big difference in values?

Response 3: Thank you for your question, we are sorry to make you have such doubts. Because in the process of model establishment, the structure of the ball screw feed system is greatly simplified. We consider the thermal conductivity as a variable related to the simplified ball screw feed system, and use an optimization algorithm to solve it to adapt the thermal conductivity to the simplification of the geometry, thereby improving the prediction accuracy of the model.

Point 4: Define the main variables involved in experimentation and simulation. Screw material and its coefficient of thermal conductivity, total length L of the screw, number N of micro-units, dx value, ambient temperature, etc.

Response 4: Thanks for your suggestion, the table has been supplemented in Section 4 to define main variables involved in experimentation and simulation.

Point 5: Is the CNC system used in the experiment open source or reconfigurable? How is the estimated thermal error fed back into the control loop?

Response 5: In the experiment, the secondary development platform of Huazhong 8 CNC system is used. This platform provides a complete secondary development interface, which can be used for us to complete the expansion development of the thermal error compensation function. Calculate and refresh the thermal error compensation table in the HMI, and the refresh frequency is 5Hz. The thermal error compensation table is read in the NCK, the reading frequency is 1000Hz, the compensation value is generated in real time, and it is added as an offset to the control position calculated by the interpolation of the CNC system in real time, and finally transmitted to the servo system.

Reviewer 3 Report

The manuscript entitled, "Novel real-time compensation method for machine tool’s ball screw thermal error" describes the development of a heat flow model for a 3-axis milling machine that can provide thermal error correction to the location of the bit during milling.  Overall, I found the paper to be very well done and I thought the approach that was taken to develop the real-time model to be quite interesting.  I did have a couple of concerns that I hope the authors could address:

1. From Table 2: Wouldn't K1, K2, K3, and K4 be functions of the bit's rotation rate - rather than constants?  Or is the mill operated only at a constant rotation rate? 

2. Figure 5 - It's not clear to me from the images what the laser is measuring.  Is the instrument moved around during the experiments?  I think some labels and an explanation of how the measurements were performed would be helpful.

3. Figures 6 and 7 - What is the source for the periodicity in the x and y-errors?

4. Figure 9 - What do HMI and NCK mean?

5. figure 11 - Are the peaks and valleys in the error-compensated positions due to the change-over in feed speed - or is something else going on?

6. It looks like equation 10 is using a forward euler discretization method to solve the heat equation.  However, this method can become unstable if dt becomes too large or dx to small.  How does the algorithm avoid instability? 

7.  What kind of hardware do the control programs run on?  Is the sampling rate so much higher than the feed rate that instabilities in the numerical solution are not a concern?  

Author Response

Response to Reviewer 3 Comments

Point 1: From Table 2: Wouldn't K1, K2, K3, and K4 be functions of the bit's rotation rate - rather than constants?  Or is the mill operated only at a constant rotation rate?

Response 1: Thank you for your question, I am sorry to make you have such doubts. In this paper, in order to simplify the calculation of the model and improve the calculation efficiency, the convective heat transfer coefficient, heat generation at the nut, and heat generation at the bearing are simplified as being linearly related to the screw speed. k1, k2, k3, and k4 are parameters used to construct these linear relationships. In this paper, they are considered as constants for a single ball screw feed system. The feed speed of the heat engine in different experimental groups is different, 3000mm/min, 6000mm/min, 9000mm/min respectively.

Point 2: Figure 5 - It's not clear to me from the images what the laser is measuring.  Is the instrument moved around during the experiments?  I think some labels and an explanation of how the measurements were performed would be helpful.

Response 2: Thanks for your suggestion, the schematic diagram of laser interferometer measurement has been supplemented. The laser measures the distance between the receiver attached to the spindle and the emitter attached to the machine table. To measure distances in different directions, laser interferometers need to be set up in different ways. During the experiment, the instrument moves with the thermal movement of the machine tool (it will not manually adjust its erection mode during a single experiment). At different stages of the heat engine, the machine tool is run to a series of fixed measuring points, and the thermal error at the corresponding measuring point is obtained by comparing the distance measured at the same measuring point with the distance measured when the machine tool is in a cold state. A single experiment can only measure the thermal error in one direction with the laser interferometer.

Point 3: Figures 6 and 7 - What is the source for the periodicity in the x and y-errors?

Response 3: The error comes from the thermal elongation of the lead screw, and the periodicity of the error comes from the set experimental conditions. Since the experimental conditions we designed use the first heat engine for 84 minutes, then the cold engine for 42 minutes, and the last second heat engine for 42 minutes (Table 1). Therefore, the lead screw first generates thermal elongation, then retracts, and then generates thermal elongation again, so the thermal error Over time, it becomes larger first, then smaller, and then larger again, which is consistent with our loading conditions.

Point 4: Figure 9 - What do HMI and NCK mean?

Response 4: HMI is the abbreviation of Human Machine Interface, which is refreshed every 200ms and has a low refresh rate. It runs on the upper layer of the CNC system for human-computer interaction and to complete some low-real-time calculations. NCK is the abbreviation of Numerical Control Kernel, which is refreshed every 1ms with a high refresh rate, and runs on the bottom layer of the CNC system to precisely control the movement of the machine tool.

Point 5: figure 11 - Are the peaks and valleys in the error-compensated positions due to the change-over in feed speed - or is something else going on?

Response 5: Yes, it exists. The greater the feed rate, the greater the steady-state thermal error. When the feed rate changes from 9000 to 3000, the thermal error may become smaller because the thermal error value has exceeded the steady-state thermal error value when the feed rate is 3000, and a peak will be generated at this time. When the feed rate is changed from 3000 to 6000, the steady-state thermal error will become larger, and the thermal error will also become larger at this time, and a valley will be generated at this time.

Point 6: It looks like equation 10 is using a forward euler discretization method to solve the heat equation.  However, this method can become unstable if dt becomes too large or dx to small.  How does the algorithm avoid instability?

Response 6: Yes, improper values of dt and dx will make the method unstable, and the research on its stable conditions is also the follow-up direction of this research. In this study, the value of dt is 0.1s, and the value of dx is 0.01m. The algorithm is stable when dx and dt take such values through practice.

Point 7: What kind of hardware do the control programs run on?  Is the sampling rate so much higher than the feed rate that instabilities in the numerical solution are not a concern?

Response 7: The program runs directly on the Huazhong 8 CNC system. The refresh frequency of the kernel system is 1000Hz. The calculation and update of the thermal error compensation table are completed in the HMI thread with the refresh frequency of 5 Hz.The compensation command is calculated in real time according to the current lead screw position feedback as the kernel refreshes.

Round 2

Reviewer 1 Report

The comments are addressed well and utilized to improve the manuscript. The manuscript is acceptable.

Also, some details about the “MATLAB programming” and “it is implemented by C++ programming” should be mentioned in the paper, as follow:

“Point 5: What software was used for the implementation of the algorithm presented in the Figure 4 “Flowchart of screw thermal error prediction”?
Response 5: In the stage of parameter identification, it is implemented by matlab programming, and the parameters to be identified are calculated by solving the optimization equation in matlab. In the stage of thermal error compensation, based on the secondary development platform of Huazhong 8 CNC system, it is implemented by C++ programming, and the thermal error compensation value is calculated in real time. In the future, the parameter identification is ready to be applied in the Huazhong 9 CNC system and realized as a product.”

Author Response

Response to Reviewer 1 Comments

Point 1: Also, some details about the “MATLAB programming” and “it is implemented by C++ programming” should be mentioned in the paper, as follow:

“Point 5: What software was used for the implementation of the algorithm presented in the Figure 4 “Flowchart of screw thermal error prediction”?

Response 5: In the stage of parameter identification, it is implemented by matlab programming, and the parameters to be identified are calculated by solving the optimization equation in matlab. In the stage of thermal error compensation, based on the secondary development platform of Huazhong 8 CNC system, it is implemented by C++ programming, and the thermal error compensation value is calculated in real time. In the future, the parameter identification is ready to be applied in the Huazhong 9 CNC system and realized as a product.”

Response 1: Thank you for your suggestion, the relevant parts have been supplemented in the revised manuscript, as: “The algorithm is implemented by matlab in the parameter identification process, and implemented in the HNC-848 CNC system by C++ in the error compensation process.” in Section 3, “The solution of the optimization equation in the process of parameter identification is completed by matlab.” in Section 4.2, and “Based on the above model, a function for the compensation of screw thermal error was integrated into the HNC-848 CNC system by C++ programming” in Section 4.3.
